# Evaluation of Essential Oils as Sprout Suppressants for Potato (*Solanum tuberosum*) at Room Temperature Storage

**DOI:** 10.3390/plants11223055

**Published:** 2022-11-11

**Authors:** Jena L. Thoma, Charles L. Cantrell, Valtcho D. Zheljazkov

**Affiliations:** 1Department of Crop and Soil Science, Oregon State University, 109 Crop Science Building, 3050 SW Campus Way, Corvallis, OR 97331, USA; 2Natural Products Utilization Research Unit, Agricultural Research Service, United States Department of Agriculture, University, MS 38677, USA

**Keywords:** potato storage, essential oils, sprout suppression, organic agriculture, room temperature

## Abstract

Chlorpropham (CIPC) has been the dominant method of chemical sprout suppression for the last half-century. However, stricter regulations including outright bans on its use in several countries has prompted investigation into alternative products to replace it. Growing interest in organic foods has increased focus on the use of biopesticides, including essential oils (EOs), as potential sprout suppressants in stored potato. We evaluated the potential of ten EOs for sprout suppression in potato cultivar Ranger Russet at room temperature. Treatment with *Cymbopogon citratus* EO was found to be the most effective sprout suppressant, completely suppressing sprouting over the 90-day storage period. The EOs of *Myrtus communis* and *Melaleuca quinquenervia* significantly reduced sprout length relative to the control but did not have any effect on sprout number. These findings demonstrate the potential of select EOs as effective potato sprout suppressants that could replace CIPC use in this industry while also giving more power to organic potato producers and processors to control sprouting in their operations.

## 1. Introduction

Potato (*Solanum tuberosum*) is the fourth largest crop after maize, wheat, and rice, and is cultivated in over 100 countries [1]. The tubers are a staple in the diets of numerous cultures and are highly versatile in cooking; fresh potatoes can be prepared in many ways or processed into a variety of products [2]. Globally, over 370 million tons of potato were produced in 2019, and over 19 million tons were produced in the United States alone [3]. Sales of potatoes grown in the United States in 2020 totaled over $3.6 billion [4]. Potato is therefore of major economic importance in this region and on the global scale.

Potatoes must often be stored for several months before being consumed or used as seed for the establishment of the next crop [5]. Immediately following harvest, most potato cultivars are in a natural state of dormancy and will not sprout. The length of this innate dormancy is influenced by environmental, physiological, and hormonal factors, and is highly cultivar dependent [6,7]. However, even the longest periods of innate dormancy are often insufficient to meet the needs of overall markets [1]. Extended storage periods can promote increased water loss, occurrence of disease, and sprouting in tubers [8]. Control of sprouting during storage is crucial as sprouting leads to changes in tuber weight, texture, and nutritional value, and the formation of toxic alkaloids including solanine [6,9]. Sprouting thereby leads to economic losses as solanine accumulation renders the potatoes inedible and they become food waste [2].

Several strategies may be implemented to limit potato sprouting during storage. These include storage at low temperatures and the use of chemical sprout suppressants. Long-term storage (up to 9 months) at temperatures between 8–12 °C at 85–90% relative humidity is a common approach to preserve processing potato quality [5]. However, once the innate dormancy period of tubers has ended, these temperatures are not sufficient on their own to prevent sprouting and sprout elongation [5]. While storage at lower temperatures is possible, cold temperatures increase glucose concentrations in the tuber flesh which causes unacceptable darkening in processed potato products and the formation of acrylamide during frying, a compound that may pose health risks to humans [10,11]. The costs of installing, running, and maintaining cooling systems may also be prohibitive, particularly to small and medium facility owners [12]. Chemical sprout suppressants, on the other hand, may offer an effective and less expensive approach to achieving sprout suppression.

The potato industry has relied heavily on the use of chemical sprout suppressants such as chlorpropham (CIPC) since the mid-20th century [1]. CIPC is an inexpensive and highly effective sprout inhibitor that interferes with mitosis in developing bud tissue, requiring only a single application to achieve complete sprout suppression for up to 5 months [13,14]. However, growing concerns about potentially adverse health and environmental effects of CIPC and its metabolites has led to stricter limitations on allowable residues and an outright ban on CIPC in the European Union [15,16]. The total value of exported potatoes from the US in 2017 was around $3 billion [17]. With the ban of CIPC in the EU and other countries, US potato exports will suffer if American-grown potatoes cannot be sold to countries with zero-tolerance policies for CIPC residues. Meanwhile, a global market for organically produced foods and products has grown significantly in recent years; organic sales totaled $62 billion in the United States alone in 2021 [18]. Taken together, the new laws on CIPC use and growing interest in organic foods indicate high potential for expanded use of alternative chemical sprout suppressants such as essential oils (EOs) in potato sprout suppression. Doing so would not only reduce the economic impact that these bans have on the US potato industry but would also give organic potato growers and processors more control over sprouting in their operations.

Several EO-containing sprout suppressants are currently available including Biox-M, Biox-C, and Talent^®^ [1]. Biox-M, containing 100% spearmint (*Mentha spicata* L.) EO, is the most common EO sprout suppressant used in the United States, accounting for 3% of treatments to stored potatoes in 2016 [16]. Biox-C contains 100% clove EO (*Syzygium aromaticum* L.), whereas Talent^®^ contains caraway EO (*Carum carvi* L.) [1]. However, the efficacy of spearmint, clove, and caraway EOs on sprout suppression at room temperature may be inconsistent or inadequate and can vary with cultivar [12,16,19,20,21]. Furthermore, there may be other EOs with sprout suppressive qualities yet to be identified. This study was conducted to evaluate the effect of ten previously untested EOs on sprouting in one potato cultivar, Ranger Russet, with the objective of identifying those suitable for potato storage at room temperature.

## 2. Results and Discussion

### 2.1. Effects of Essential Oils (EOs) on Longest Sprout Length

After 90 days of storage, a statistically significant two-way interaction between treatment and time was observed on sprout length (Table 1). This suggests that the impact of treatment on sprout length depends on the amount of time that has passed. Furthermore, the main effects of both treatment and time were significant (Table 1).

The EOs of *Myrtus communis*, *Melaleuca quinquenervia*, and *Cymbopogon citratus* resulted in significant differences in sprout length relative to the control (Table 2). Treatment with *C. citratus* EO resulted in significant differences in sprout length from the control at all time points, whereas the effects of *M. communis* and *M. quinquenervia* EO treatments were significant between 45–90 days and 45–75 days of storage, respectively (Table 2).

Complete suppression of sprouting was obtained over the entire 90-day storage period with *C. citratus* EO treatment (Figure 1). Sprout length due to *C. citratus* EO treatment differed significantly from that due to either *M. communis* or *M. quinquenervia* EO treatment from 45 days until the end of the storage period (*p* < 0.001, Tukey’s test) (Table 3). Sprout length due to *M. communis* EO treatment did not differ from that of *M. quinquenervia* EO treatment at any time point (*p* > 0.05, Tukey’s test) (Table 3). Repeated applications of EOs are often required to maintain adequate sprout suppression over longer storage periods [16,22,23]. Perhaps more effective sprout suppression could be achieved with *M. communis* and *M. quinquenervia* EOs if repeated applications were used.

*Myrtus communis* is a medicinal plant used in the food, pharmaceutical, and cosmetic industries [24]. GC analysis of *M. communis* EO reveals α-pinene, eucalyptol, linalool, p-cymene, geranyl acetate, and α-terpineol as major constituents (Table 4). Geranyl acetate and α-pinene have previously been reported as an effective and somewhat effective sprout suppressant, respectively [1]. Furthermore, these monoterpenes have also been shown to inhibit germination and growth in other species, perhaps through their induction of oxidative stress [25,26,27]. It is possible that these compounds are responsible for this EO’s sprout suppressive properties. *M. communis* EO has been associated with decreased weight and fruit firmness losses in strawberries [28]. Furthermore, sunflower oil fortified with *M. communis* EO can enhance the physiochemical properties of potato chips and could significantly increase their shelf-life [29]. These reports and the findings of the current study suggest that *M. communis* EO could be an effective component in EO sprout suppressant formulations. 

*Melaleuca quinquenervia* is a medicinal plant with applications in aromatherapy, cosmetics, and pharmaceuticals, although its use in the food industry has been suggested [34]. GC analysis reports eucalyptol, α-terpineol, α-pinene, viridiflorol, β-pinene, caryophyllene, viridiflorene, α-terpinyl acetate, and p-cymene as major constituents (Table 5). Of these compounds, only α-pinene has been reported as a somewhat effective sprout suppressant [1]. It is possible that the other major compounds, perhaps in combination with α-pinene, are responsible for the sprout suppressive capabilities of this EO. The findings of the current study encourage further investigation of both *M. quinquenervia* and *M. communis* EOs as sprout suppressants in other cultivars and using different application schemes, such as repeated or continuous application, to achieve more effective sprout control.

*Cymbopogon citratus* is another medicinal plant commonly used in the pharmaceutical and cosmetic industries with potential applications in the food industry [37]. *C. citratus* EO is previously reported to be high in citral, a compound associated with effective sprout suppression in potato [1,37]. GC analysis of *C. citratus* EO used in this study confirms the presence of citral, an aldehyde mixture of neral and geranial. Major compounds of *C. citratus* EO include geranial, neral, geraniol, geranyl acetate, 6-methyl-5-heptene-2-one, camphene, linalool, 4-nonanone, γ-cadinene, caryophyllene, and limonene (Table 6). In addition to geranial and neral, geranyl acetate has also been reported as an effective sprout suppressant [1]. Given their high proportions within *C. citratus* EO, it is likely that these compounds are responsible for the complete suppression of sprouting observed in this study.

Owolabi et al. [38] demonstrated the potential of *C. citratus* EO as a sprout suppressant in Russet Burbank potatoes, corroborating the findings in the present study. Furthermore, Belay et al. [39] reported lower weight loss over a 14-week storage period in tubers treated with *C. citratus* EO, however, no difference in sprout length relative to a control was observed in either cultivar tested. EOs are known to show differences in sprout suppression depending on potato cultivar [16,23,39]. Therefore, it is possible that *C. citratus* EO is more effective in cultivars Ranger Russet, used in the present study, and Russet Burbank than it is in Gudene or Jalene [38,39]. *C. citratus* EO application has also been associated with reductions in potato tuber moth (*Phthorimaea opperculella*) infestation, suggesting that its use could provide additional benefits besides sprout suppression [40].

### 2.2. Effects of Essential Oils (EOs) on Number of Germinated Eyes

After 90 days of storage, a statistically significant two-way interaction between treatment and time was observed on sprout number (Table 1). This suggests that the impact of treatment on sprout number depends on the amount of time that has passed. Furthermore, the main effects of both treatment and time were significant (Table 1).

Only treatment with *C. citratus* EO resulted in a significant difference in sprout number relative to the control (Table 7). Though not significant, treatment with *Pogostemon cablin* EO resulted in slightly fewer sprouts relative to the control from 60–90 days (Figure 2). Complete suppression of sprouting was obtained over the entire 90-day storage period with *C. citratus* EO treatment, resulting in zero sprouting throughout the study (Figure 2). This contradicts findings by Belay et al. [39] reporting no effect of *C. citratus* EO on sprout numbers over the course of 16 weeks in cultivars Jalene and Gudene. However, like effects on sprout length, EO treatment effects on sprout number vary with cultivar [39]. The results of the present study suggest that *C. citratus* is a particularly effective sprout suppressant in Ranger Russet potatoes at room temperature.

Essential oils (EOs) are generally believed to achieve sprout suppression by damaging the meristematic tissue of developing sprouts [22]. Indeed, 1,8-cineole application has been associated with complete necrosis of potato sprout tissue, whereas α-pinene and citral, an aldehyde mixture of geranial and neral, have been associated with necrosis of just the sprout tips [41]. However, sprouting was completely inhibited with *C. citratus* EO treatment, and no visible sprouts were observed throughout the 90-day storage period (Figure 3). SEM of tuber eyes from *C. citratus* treated potatoes reveals healthy, undeveloped bud tissue (Figure 4). Due to lack of discernable damage to the tuber bud tissue, it is therefore possible that *C. citratus* treatment may achieve sprout suppression through a mechanism that is different from the physical damage typically associated with EO sprout suppressants. For example, it is possible that *C. citratus* EO treatment may suppress sprouting by interfering with the plant hormone balance within the tubers or through another mechanism entirely. Several plant hormones including auxins, gibberellins, cytokinins, and abscisic acid are known to regulate dormancy release and sprouting in potato tubers [42]. Previous studies suggest that compounds such as citral may play a role in gibberellin and indole-3-acetic acid suppression in potato [43], whereas pinene isomers have been associated with fluctuations in abscisic acid concentrations [25]. Furthermore, 1,8-cineole and α-pinene have been shown to alter mitochondrial metabolism in corn, inhibiting root growth and interfering with germination [44] Similar mechanisms may be present in potato. Additionally, garlic EO application on potato tubers has been shown to alter the abundance of specific proteins associated with seed germination [45]. It is therefore possible that EOs achieve sprout suppression through a variety of mechanisms. Further studies performing proteomic analysis of EO-treated potatoes or investigating the effect of *C. citratus* EO and its pure components on the levels of various plant hormones in treated potatoes could help to determine the active ingredient(s) and provide important insights into their mode of action. This could hasten the identification of additional EOs with sprout suppressive capabilities and identify new target genes in tuber breeding programs.

Recent studies suggest that various *Cymbopogon spp.* EOs may be promising potato sprout manipulators. *Cymbopogon martini* EO has been associated with sprout suppression at temperatures above 20 °C [12,39]. Similarly, *Cymbopogon nardus* EO can completely suppress sprouting for up to 30 days after dormancy break [46]. Conversely, *Cymbopogon schoenanthus* EO has been associated with sprout enhancement and increased yields [12]. The present study suggests that *C. citratus* EO is also an effective sprout suppressant at room temperature, corroborating previous findings by Owolabi et al. [38]. Greater focus and investigation into other species in this genus as potato sprout modulators in a wider range of cultivars is thus warranted. 

The EO composition can vary widely depending on the plant parts used for extraction [35]. For this reason, it is possible that EOs not observed to be effective sprout suppressants in the present study could possess sprout suppressive properties if different plant parts are used for extraction. Similarly, different extractions of EOs shown to be effective in the present study from one or more plant parts may display variable effects on potato sprouting due to varying compositions. Indeed, studies comparing the effects of EOs extracted from the bark, leaves, or fruit of these species on potato sprouting could more fully illustrate their potential as sprout suppressants while expediting the identification of active ingredients and the best sources of EOs for use in this industry. Nevertheless, the EO of *C. citratus*, the most potent treatment, is commonly extracted from the whole aboveground plant parts in vegetative stage, that is stems and leaves. 

Potatoes may be stored for many months before use, requiring multiple applications of sprout suppressants [13]. As only a single application of each EO was used, and typically, EO-based products are applied every 2 or 4 weeks, the storage period in the present study was set to 90 days. Future studies could investigate longer storage lengths or may test different concentrations of the most effective EOs. Importantly, even if an EO possesses significant sprout suppressive or inhibitory properties, it is possible that its application may alter the flavor, texture, or nutritional quality of treated potatoes. Therefore, additional studies are needed to investigate *C. citratus* and other EOs for their effects on other aspects of potato quality before commercial products may be developed.

Furthermore, larger studies comparing the effectiveness of *C. citratus* EO to conventional methods such as CIPC in semi-commercial and commercial settings are also needed. These types of studies are necessary to evaluate the efficacy of *C. citratus* EO in industrial settings and determine the feasibility of scaling up its use to a commercial scale.

## 3. Materials and Methods

### 3.1. Plant Material

Potato tubers of cultivar Ranger Russet were obtained from Oregon State University Hermiston Agricultural Research and Extension Center in Hermiston, OR, USA. Tubers were harvested in September 2021 and subsequently stored at 4 °C. All tubers were left untreated by any chemicals prior to the start of the experiment. Container studies were initiated in November 2021.

### 3.2. Experimental Materials

A total of 10 essential oils (EOs) including myrtle (*Myrtus communis*), niaouli (*Melaleuca quinquenervia*), nutmeg (*Myristica fragrans*), opopanax (*Commiphora erythraea*), bitter orange (*Citrus aurantium*), sweet orange (*Citrus sinensis*), palo santo (*Bursera graveolens*), parsley seed (*Petroselinum sativum*), patchouli (*Pogostemon cablin*), and lemongrass (*Cymbopogon citratus*) were used (Table 8). *Cymbopogon citratus* EO was purchased from Greenway Biotech, Inc. All other EOs were purchased from Mountain Rose Herbs (Eugene, OR, USA).

### 3.3. Experimental Design

One (1) mL of EO was pipetted on to a cotton ball sitting in a glass Petri dish lined with filter paper in the center of a new, previously unused black 20 L plastic container. Three randomly selected tubers were placed in each container and the containers were then sealed with aluminum foil for fumigation with the EO vapor. The Petri dish with EO had no direct contact with the tubers. A loose-fitting lid was placed on the containers, which were then stacked and left undisturbed aside from scheduled intervals for data collection as described in 2.4. Observations. There were 3 replications per EO treatment and the control with 1 mL distilled water, for a total of 3 mL of each EO and distilled water used in the experiment. The experiment was conducted at room temperature and lasted 90 days.

### 3.4. Observations

The effects of the treatments were evaluated by recording data on sprout length and number of sprouts starting at 30 days and continuing every 15 days thereafter until a 90-day storage period was reached.

At all data collection time points, the longest sprout on each tuber in each replication was recorded in millimeters and reported as sprout length. The total number of germinated (≥1 mm) eyes was recorded for all tubers in each replication and reported as sprout number. Averages of observations for each replication were calculated for later analysis.

### 3.5. Statistical Analysis

R software, Version 3.6.3, was used for the statistical analysis [47]. A linear mixed model was used to analyze both sprout length and sprout number. Due to wide variability of the data and to fulfill the ANOVA assumptions, a square root transformation was used on the sprout length data to achieve homogeneity of variance and normality of residuals. A post hoc Tukey’s HSD test was used as a multiple comparison test to identify differences in sprout length and number due to the different treatments across all time points. For the sprout length data, estimated marginal means and confidence intervals were back-transformed for reporting and graphics. To perform the aforementioned analysis, data summary, and graphics, we used various R packages (“ggpubr” [48], “tidyverse”, “rstatix”, “nlme”, “emmeans”, and “ggplot2” [48,49,50,51,52]).

### 3.6. Gas Chromatography Mass Spectrometry Flame Ionization Detection (GC–MS–FID) Essential Oil Analysis 

Gas chromatography (GC)–mass spectroscopy (MS)–flame ionization detection (FID) analysis of *C. citratus*, *M*. *quinquenervia* and *M. communis* EOs was performed at the Natural Products Center of the USDA-ARS, Natural Products Utilization Research Unit in University, MS, USA. Using a micropipette, 50 μL of oil (weight measured on a tared balance) from each sample was transferred into a 10 mL volumetric flask. Samples were brought to volume with CHCl_3_. A 1 mL aliquot of each diluted oil sample was placed by glass pipet into a GC vial for analysis. 

Oil samples were analyzed by GC–MS–FID on an Agilent (Santa Clara, CA, USA) 7890A GC system coupled to an Agilent 5975C inert XL MSD. Chemical standards and oils were analyzed using a DB-5 column (30 m × 0.25 mm fused silica capillary column, film thickness of 0.25 µm) operated using an injector temp of 240 °C, column temperature of 60 to 240 °C at 3 °C/min and held at 240 °C for 5 min, helium as the carrier gas, an injection volume of 1 µL (split ratio 25:1), and an MS mass range from 50 to 550. FID temperature was 300 °C. Post-column splitting was performed so that 50% of outlet flow proceeded to FID and 50% to mass spectrometry (MS) detection. 

Compounds were identified by Kovats Index analyses, direct comparison of MS and retention time to authentic standards, and comparison of mass spectra with those reported in the Adams and NIST mass spectra databases, unless otherwise noted. Commercial standards were obtained from Sigma-Aldrich (St. Louis, MO, USA) for direct comparison. 

Compounds were quantified by performing area percentage calculations based on the total combined FID area. For example, the area for each reported peak was divided by total integrated area from the FID chromatogram from all reported peaks and multiplied by 100 to arrive at a percentage. The percentage of a peak is a percentage relative to all other constituents integrated in the FID chromatogram.

### 3.7. Scanning Electron Microscopy

Three eyes were randomly selected and removed from whole tubers treated with *C. citratus* EO and placed in micro-centrifuge tubes containing deionized water. The eyes were ultra-sonicated for 5 min to remove remaining soil. The wash water was then removed and chemical fixative (2.5% glutaraldehyde, 1% paraformaldehyde in 0.1 M sodium cacodylate buffer) was added. The potato eyes remained in the fixative for 60 h at 40 °F. Samples were rinsed and serial dehydrated with ethanol and critical point dried for scanning electron microscopy (SEM). Dried samples were attached to a SEM stub mount with carbon tape and sputter coated with gold-palladium. Images were acquired using a Quanta 600 FEG SEM at Oregon State University.

## 4. Conclusions

Growing interest in organic foods and stricter regulations on the use of CIPC make EO sprout suppressants uniquely poised for expanded use in potato sprout suppression. While several EO-containing sprout suppressants are currently available, they exhibit variable efficacy depending on storage conditions, potato cultivars, and application schemes. The wide variability in plant secondary metabolites and in the chemical profiles of EOs suggests that many additional, effective EO sprout suppressants have yet to be discovered. The present study identified *C. citratus* EO as a highly effective sprout suppressant that completely suppresses sprouting in Ranger Russet potatoes for up to 90 days at room temperature storage. Results also suggest that the efficacy of *M. communis* and *M. quinquenervia* EOs could be enhanced if they are applied repeatedly during storage. This study clearly demonstrated the ability of select EOs to control sprouting in stored potato, offering an organic alternative to present practices dependent on the use of CIPC. Application schemes of these EOs in commercial settings will need to be investigated and optimized and their modes of action can be explored. Doing so will realize the benefits of their use in the potato industry and could allow for identification of other promising sprout suppressants via composition alone.

## Figures and Tables

**Figure 1 plants-11-03055-f001:**
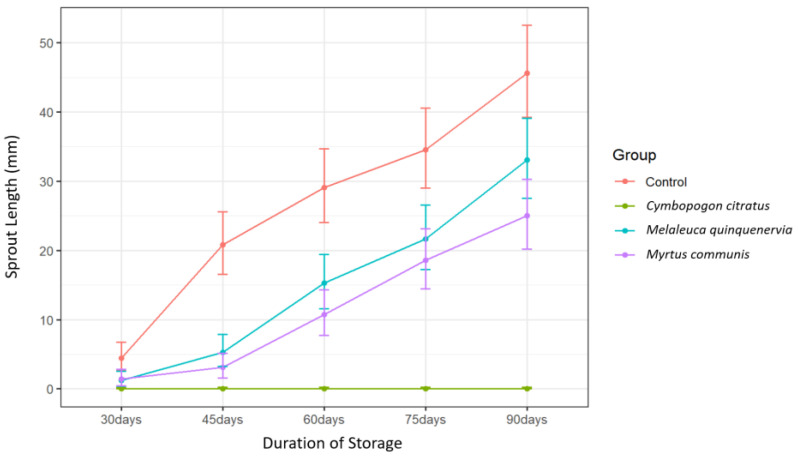
Sprout length (mm) over time of potatoes treated with distilled water (control), *Cymbopogon citratus*, *Myrtus communis*, and *Melaleuca quinquenervia* EOs. Error bars represent the 95% confidence level of the back-transformed means (*emmeans method*).

**Figure 2 plants-11-03055-f002:**
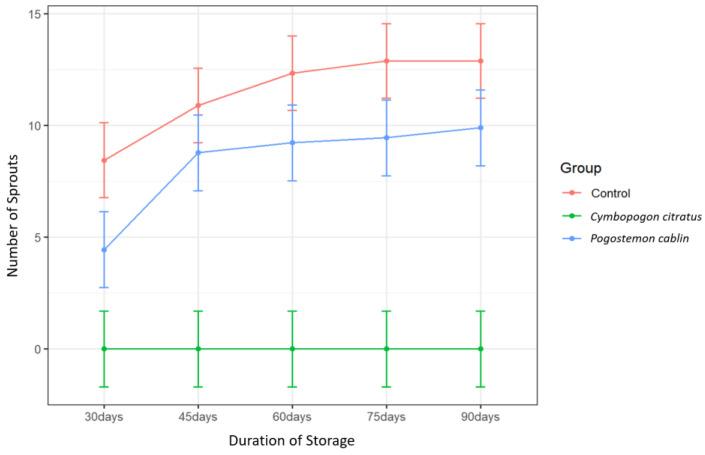
Number of sprouts per tuber over time of potatoes treated with distilled water (control), *Cymbopogon citratus*, and *Pogostemon cablin* EOs. Error bars represent the 95% confidence level of the means (*emmeans method*).

**Figure 3 plants-11-03055-f003:**
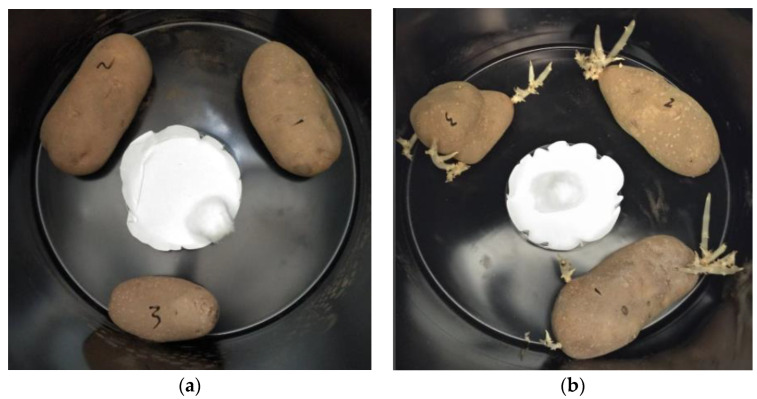
Ranger Russet tubers treated with (**a**) *Cymbopogon citratus* essential oil (EO) and (**b**) distilled water at 90 days of storage.

**Figure 4 plants-11-03055-f004:**
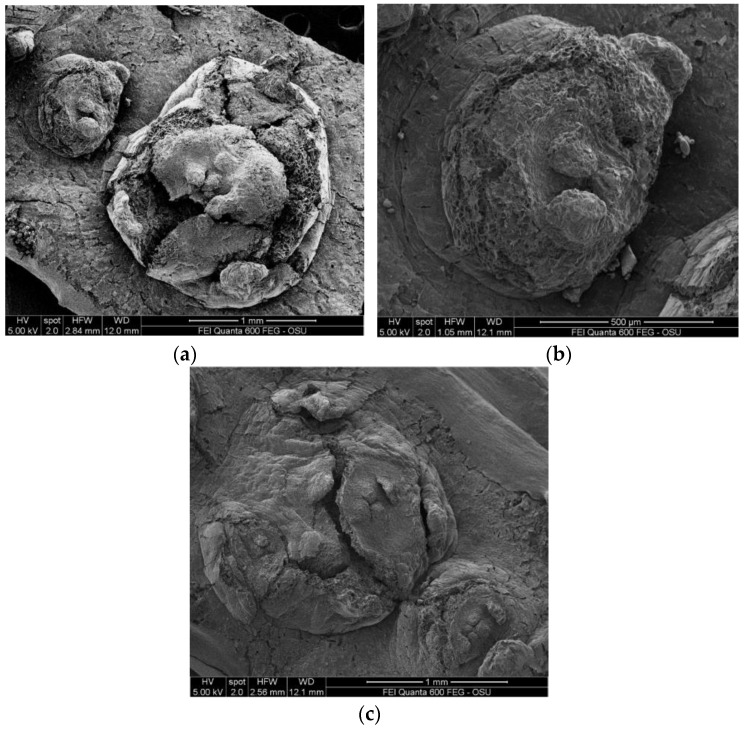
Scanning electron microscopy images of three potato eyes (**a**–**c**) from tubers treated with *Cymbopogon citratus* EO after 90 days of storage.

**Table 1 plants-11-03055-t001:** ANOVA *p*-values (*Pr[>F]*) of the linear mixed model for sprout length and number in response to treatment and time and their interactions (*** = statistically significant).

Variables	Sprout Length	Sprout Number
Treatment	<0.001 ***	<0.001 ***
Time	<0.001 ***	<0.001 ***
Treatment × Time	<0.001 ***	<0.001 ***

**Table 2 plants-11-03055-t002:** Tukey’s test *p*-values describing the EOs’ effects on sprout length relative to the control at all time points (*, **, *** = statistically significant).

Plant EO	Duration of Storage				
	30 Days	45 Days	60 Days	75 Days	90 Days
*Myrtus communis*	0.3310	<0.001 ***	<0.001 ***	0.00246 **	0.0257 *
*Melaleuca quinquenervia*	0.2200	<0.001 ***	<0.001 ***	0.0286 *	0.513
*Myristica fragrans*	0.9680	0.098	0.387	0.920	0.984
*Commiphora erythraea*	0.9990	0.985	1.000	1.000	0.999
*Citrus aurantium*	1.0000	0.967	1.000	1.000	0.999
*Citrus sinensis*	0.9910	1.000	0.998	0.998	1.000
*Bursera graveolens*	1.0000	0.999	1.000	1.000	0.997
*Petroselinum sativum*	1.0000	0.590	0.300	0.313	0.221
*Pogostemon cablin*	1.0000	0.265	0.356	0.891	0.796
*Cymbopogon citratus*	<0.001 ***	<0.001 ***	<0.001 ***	<0.001 ***	<0.001 ***

**Table 3 plants-11-03055-t003:** Longest sprout length (mm) of potato tubers treated with different essential oils at different time points.

	Duration of Storage				
	30 Days	45 Days	60 Days	75 Days	90 Days
Control	4.46 ± 1.07 ^a^	20.83 ± 2.32 ^a^	29.09 ± 2.74 ^a^	34.55 ± 2.99 ^a^	45.63 ± 3.43 ^a^
*Myrtus communis*	1.42 ± 0.60 ^ac^	3.11 ± 0.89 ^c^	10.77 ± 1.67 ^c^	18.55 ± 2.19 ^b^	24.99 ± 2.54 ^b^
*Melaleuca quinquenervia*	1.20 ± 0.55 ^ac^	5.29 ± 1.17 ^c^	15.28 ± 1.98 ^bc^	21.66 ± 2.36 ^bc^	33.07 ± 2.92 ^ab^
*Myristica fragrans*	2.75 ± 0.84 ^a^	13.11 ± 1.84 ^ab^	22.84 ± 2.43 ^ab^	29.32 ± 2.75 ^ab^	38.99 ± 3.17 ^a^
*Commiphora erythraea*	5.70 ± 1.21 ^a^	17.97 ± 2.15 ^ab^	29.15 ± 2.74 ^a^	32.87 ± 2.91 ^ac^	41.09 ± 3.26 ^a^
*Citrus aurantium*	3.70 ± 0.97 ^a^	17.63 ± 2.13 ^ab^	27.77 ± 2.68 ^a^	32.64 ± 2.90 ^ac^	41.00 ± 3.25 ^a^
*Citrus sinensis*	6.19 ± 1.26 ^ab^	21.32 ± 2.35 ^ab^	31.50 ± 2.85 ^a^	37.87 ± 3.13 ^a^	42.32 ± 3.31 ^a^
*Bursera graveolens*	4.92 ± 1.12 ^a^	18.79 ± 2.20 ^ab^	28.55 ± 2.71 ^a^	32.53 ± 2.90 ^ac^	40.19 ± 3.22 ^a^
*Petroselinum sativum*	4.17 ± 1.04 ^a^	15.63 ± 2.01 ^ab^	22.42 ± 2.41 ^ab^	25.53 ± 2.57 ^ab^	30.13 ± 2.79 ^a^
*Pogostemon cablin*	3.50 ± 0.95 ^a^	14.29 ± 1.92 ^ab^	22.70 ± 2.42 ^ab^	29.03 ± 2.74 ^ab^	35.58 ± 3.03 ^a^
*Cymbopogon citratus*	0 ^c^	0 ^d^	0 ^d^	0 ^d^	0 ^c^

Values are the back-transformed means ± SE (emmeans method). Different letters (^a–d^) within columns indicate statistically significant differences between treatments (Tukey’s test *p* < 0.05).

**Table 4 plants-11-03055-t004:** *Myrtus communis* EO constituents identified via GC–MS–FID analysis.

No.	Compound Name	Retention Time	Calculated KI	Actual KI	Identified	Area %
1	isobutyl isobutyrate	5.414	909	911	Kovat, NIST, Adams, [30,31,32]	0.495
2	α-thujene	5.82	926	930	Kovat, NIST, Adams, [30,31,32]	0.354
3	α-pinene	6.089	936	939	Kovat, NIST, Adams, Commercial Standard	49.086
4	β-pinene	7.23	975	979	Kovat, NIST, Adams, Commercial Standard	0.4
5	unknown	7.954	-	-	-	0.47
6	3-carene	8.27	1006	1011	Kovat, NIST, Adams, Commercial Standard	0.195
7	unknown	8.394	-	-	-	0.185
8	p-cymene	8.768	1023	1024	Kovat, NIST, Adams, Commercial Standard	2.123
9	eucalyptol	9.069	1032	1031	Kovat, NIST, Adams, Commercial Standard	33.119
10	α-pinene oxide	11.493	1099	1099	Kovat, NIST, Adams, Commercial Standard	0.664
11	linalool	11.584	1101	1096	Kovat, NIST, Adams, Commercial Standard	2.266
12	unknown	11.666	-	-	-	0.582
13	unknown	11.866	-	-	-	0.891
14	α-campholenal	12.575	1126	1126	Kovat, NIST, Adams, [33]	0.22
15	trans-pinocarveol	13.168	1141	1139	Kovat, NIST, Adams, [31,33]	0.48
16	trans-verbenol	13.431	1147	1144	Kovat, NIST, Adams, Commercial Standard	0.866
17	terpinen-4-ol	14.749	1177	1177	Kovat, NIST, Adams, Commercial Standard	0.193
18	α-terpineol	15.353	1189	1188	Kovat, NIST, Adams, Commercial Standard	1.215
19	verbenone	16.061	1205	1205	Kovat, NIST, Adams, Commercial Standard	0.42
20	trans-carveol	16.576	1219	1216	Kovat, NIST, Adams, Commercial Standard	0.289
21	(R)-carvone	17.523	1243	1243	Kovat, NIST, Adams, Commercial Standard	0.114
22	linalyl acetate	18.004	1255	1257	Kovat, NIST, Adams, [31,32]	0.581
23	unknown	20.026	-	-	-	1.248
24	unknown	20.714	-	-	-	0.552
25	α-terpinyl acetate	21.992	1348	1349	Kovat, NIST, Adams, Commercial Standard	0.294
26	geranyl acetate	23.416	1379	1381	Kovat, NIST, Adams, Commercial Standard	1.766
27	methyl eugenol	24.322	1398	1403	Kovat, NIST, Adams, Commercial Standard	0.321
28	caryophyllene oxide	31.426	1577	1583	Kovat, NIST, Adams, Commercial Standard	0.477
29	unknown	32.43	-	-	-	0.133

**Table 5 plants-11-03055-t005:** *Melaleuca quinquenervia* essential oil (EO) constituents identified via GC–MS–FID analysis.

No.	Compound Name	Retention Time	Calculated KI	Actual KI	Identified	Area %
1	α-thujene	5.81	926	930	Kovat, NIST, Adams, [35]	0.132
2	α-pinene	6.028	934	939	Kovat, NIST, Adams, Commercial Standard	7.081
3	camphene	6.42	948	954	Kovat, NIST, Adams, Commercial Standard	0.056
4	benzaldehyde	6.731	959	960	Kovat, NIST, Adams, Commercial Standard	0.154
5	β-pinene	7.232	975	979	Kovat, NIST, Adams, Commercial Standard	2.037
6	myrcene	7.597	986	990	Kovat, NIST, Adams, Commercial Standard	0.974
7	unknown	8.063	-	-	-	0.105
8	α-terpinene	8.482	1012	1017	Kovat, NIST, Adams, Commercial Standard	0.19
9	p-cymene	8.8	1024	1024	Kovat, NIST, Adams, Commercial Standard	1.31
10	eucalyptol	9.125	1034	1031	Kovat, NIST, Adams, Commercial Standard	64.754
11	γ-terpinene	9.976	1059	1059	Kovat, NIST, Adams, Commercial Standard	0.764
12	terpinolene	11.091	1088	1088	Kovat, NIST, Adams, Commercial Standard	0.494
13	linalool	11.561	1100	1096	Kovat, NIST, Adams, Commercial Standard	0.135
14	δ-terpineol	14.329	1167	1166	Kovat, Adams, [36]	0.177
15	terpinen-4-ol	14.753	1177	1177	Kovat, NIST, Adams, Commercial Standard	0.793
16	α-terpineol	15.409	1190	1188	Kovat, NIST, Adams, Commercial Standard	7.439
17	α-terpinyl acetate	21.998	1348	1349	Kovat, NIST, Adams, Commercial Standard	1.434
18	α-gurjunene	24.522	1402	1409	Kovat, Adams, [35,36]	0.165
19	caryophyllene	24.938	1414	1408	Kovat, NIST, Adams, Commercial Standard	1.787
20	unknown	25.796	-	-	-	0.292
21	α-humulene	26.317	1452	1454	Kovat, NIST, Adams, Commercial Standard	0.372
22	alloaromadendrene	26.611	1459	1460	Kovat, NIST, Adams, [35,36]	0.524
23	viridiflorene	28.012	1495	1496	Kovat, Adams, Commercial Standard, [35]	1.56
24	γ-cadinene	28.747	1514	1513	Kovat, NIST, Adams, Commercial Standard	0.178
25	δ-cadinene	29.1	1523	1523	Kovat, NIST, Adams, Commercial Standard	0.255
26	trans-nerolidol	30.706	1561	1563	Kovat, NIST, Adams, Commercial Standard	0.676
27	viridiflorol	31.866	1588	1592	Kovat, NIST, Adams, Commercial Standard	5.262
28	ledol	32.249	1596	1602	Kovat, NIST, Adams, Commercial Standard	0.901

**Table 6 plants-11-03055-t006:** *Cymbopogon citratus* essential oil (EO) constituents identified via GC–MS–FID analysis.

No.	Compound Name	Retention Time	Calculated KI	Actual KI	Identified	Area %
1	tricyclene	5.723	922	921	Kovat, NIST, Adams, Commercial Standard	0.3055
2	α-pinene	6.015	933	932	Kovat, NIST, Adams, Commercial Standard	0.446
3	camphene	6.429	948	946	Kovat, NIST, Adams, Commercial Standard	2.3915
4	6-methyl-5-heptene-2-one	7.435	981	981	Kovat, NIST, Adams, Commercial Standard	2.6885
5	limonene	8.899	1027	1024	Kovat, NIST, Adams, Commercial Standard	1.23
6	trans-β-ocimene	9.171	1035	1032	Kovat, NIST, Adams, Commercial Standard	0.171
7	4-nonanone	10.436	1072	-	NIST, Adams, Commercial Standard	1.5955
8	linalool	11.552	1100	1095	Kovat, NIST, Adams, Commercial Standard	1.5965
9	unknown	11.677	-	-	-	0.218
10	unknown	13.084	-	-	-	0.231
11	unknown	13.516	-	-	-	0.3575
12	citronellal	13.646	-	-	Kovat, NIST, Adams, Commercial Standard	0.521
13	unknown	14.126	-	-	-	0.4845
14	endo-borneol	14.31	1168	1165	Kovat, NIST, Adams, Commercial Standard	0.213
15	unknown	14.88	-	-	-	0.9955
16	α-terpineol	15.323	1189	1186	Kovat, NIST, Adams, Commercial Standard	0.2205
17	neral	17.53	1243	1235	Kovat, NIST, Adams, Commercial Standard	30.295
18	geraniol	18.115	1258	1249	Kovat, NIST, Adams, Commercial Standard	6.0465
19	geranial	18.85	1275	1264	Kovat, NIST, Adams, Commercial Standard	41.4925
20	geranyl acetate	23.422	1379	1379	Kovat, NIST, Adams, Commercial Standard	4.3295
21	caryophyllene	24.935	1414	1408	Kovat, NIST, Adams, Commercial Standard	1.418
22	α-humulene	26.311	1451	1452	Kovat, NIST, Adams, Commercial Standard	0.3225
23	γ-cadinene	28.748	1513	1513	Kovat, NIST, Adams	1.4695
24	δ-cadinene	29.095	1522	1522	Kovat, NIST, Adams, Commercial Standard	0.423
25	caryophyllene oxide	31.402	1577	1582	Kovat, NIST, Adams, Commercial Standard	0.537

**Table 7 plants-11-03055-t007:** Tukey’s test *p*-values describing the essential oils (EOs’) effects on sprout number relative to the control at all time points (*** = statistically significant).

Plant EO	Duration of Storage				
	30 Days	45 Days	60 Days	75 Days	90 Days
*Myrtus communis*	0.9910	1.000	1.000	1.000	1.000
*Melaleuca quinquenervia*	0.2460	1.000	1.000	1.000	1.000
*Myristica fragrans*	0.9950	0.929	0.999	1.000	1.000
*Commiphora erythraea*	1.0000	0.992	0.999	1.000	0.999
*Citrus aurantium*	1.0000	1.000	1.000	0.997	0.999
*Citrus sinensis*	1.0000	1.000	0.995	1.000	0.998
*Bursera graveolens*	0.9220	0.992	1.000	1.000	0.999
*Petroselinum sativum*	0.9750	1.000	1.000	1.000	0.999
*Pogostemon cablin*	0.1870	0.903	0.146	0.053	0.085
*Cymbopogon citratus*	<0.001 ***	<0.001 ***	<0.001 ***	<0.001 ***	<0.001 ***

**Table 8 plants-11-03055-t008:** The plant parts, their origin, and the method of extraction used to produce the essential oils (EOs) used in the present study. Information was obtained from supplier websites.

Plant EO	Plant Parts	Country of Origin	Method of Extraction
*Myrtus communis*	Leaves and Twigs	Tunisia	Steam Distillation
*Melaleuca quinquenervia*	Leaves and Twigs	Madagascar	Steam Distillation
*Myristica fragrans*	Seeds	Sri Lanka	Steam Distillation
*Commiphora erythraea*	Resin	Somalia	Steam Distillation
*Citrus aurantium*	Peels	Egypt	Steam Distillation
*Citrus sinensis*	Peels	USA	Cold Pressed
*Bursera graveolens*	Wood	Ecuador	Steam Distillation
*Petroselinum sativum*	Seeds	France	Steam Distillation
*Pogostemon cablin*	Leaves	Sri Lanka	Steam Distillation
*Cymbopogon citratus*	Leaves and Stems	Sri Lanka	Steam Distillation

## Data Availability

Not applicable.

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
