# Peer review of "Evaluation of Essential Oils as Sprout Suppressants for Potato (Solanum tuberosum) at Room Temperature Storage"

_plants, 2022, doi:10.3390/plants11223055_

Round 1

Reviewer 1 Report

This manuscript contains useful information as it is the content that plant essential oil inhibits sprouting of potato tuber when stored at room temperature. However, although this manuscript is excellent in practical terms, it lacks academic considerations. That is, there is no consideration of why EO and the main ingredient inhibit sprouting when stored at room temperature of potato tubers. Therefore, this manuscript cannot be published in Plants at this time. Please refer to the details recorded below.

 1. The authors used 10 types of EO, but the information on them is very poor. In particular, there is too little information on the three most active species. Because, in the case of Melaleuca quinquenervia essential oil, there is a very large difference in composition depending on the leaf, fruit, and branch (Ref. UNED Research Journal Vol. 13(1): e3327, June, 2021).

 2. The authors are not satisfied with the legend representation of the data. For example, in Tables 2 and 3, it would be better to change Treatment vs Control to Plant EO, and it would be better to change 30, 45... days of X axis to Storage period for readability. Also Fig. In 2, you need to change Number to Number of sprouts and Time to Storage period.

And scientific names are in italics. need to be corrected.

3. There is no explanation as to why the storage period is set to 90 days. It is unknown whether the authors confirmed the fumigation effect of EO for 90 days prior to this experiment. As shown in Figure 1, after 45 days, the growth of the shoots accelerates rapidly. In addition, the treatment concentration of EO should be performed as a prior study. The duration of the experiment may have been short.

 4. The authors performed GC-MS analysis of the three essential oils with the best inhibitory effect. Therefore, α-pinene was found in M. communis EO, eucalyptol in M. quinquenervia, and geranial in C. citratus. However, there is no explanation as to how these components affect shoot inhibition (including previous studies). That is, the biggest problem of this paper is that there is no consideration on the germination inhibitory effect of the EO component. For example, these EO components may have factors such as inhibition of biosynthesis of hormones such as auxin, increase of biosynthesis of hormones such as ABA, and changes in energy sources such as reducing sugars. Although this experiment has not been conducted, it may be considered through previous studies.

 5. Line 168-187 does not need to be mentioned. This is because the three types of EO revealed that there is significance, and the story is mainly developed based on this.

 6. I don't know why 3 SEM pictures are listed in Figure 4. The authors would like to show only a picture of the control and processing section, or a characteristic part.

 7. Controlling potato tubers in a controlled environment (even if tuber sprouting is suppressed) may cause changes in the tissue and nutritional quality of potatoes, and may reduce tuber weight. In other words, the authors should specify the limitations of the study.

Author Response

This manuscript contains useful information as it is the content that plant essential oil inhibits sprouting of potato tuber when stored at room temperature. However, although this manuscript is excellent in practical terms, it lacks academic considerations. That is, there is no consideration of why EO and the main ingredient inhibit sprouting when stored at room temperature of potato tubers. Therefore, this manuscript cannot be published in Plants at this time. Please refer to the details recorded below.

  1. The authors used 10 types of EO, but the information on them is very poor. In particular, there is too little information on the three most active species. Because, in the case of Melaleuca quinquenervia essential oil, there is a very large difference in composition depending on the leaf, fruit, and branch (Ref. UNED Research Journal Vol. 13(1): e3327, June, 2021).

Response: We acknowledge the importance of knowing the specific plant parts used for EO extraction. However, as all 10 of the EOs used in this study were purchased from commercial suppliers, this information is not readily available. By providing the GC analysis of the three most active species, we believe we address this concern, as the goal of the study was not to determine the best source of individual EOs in terms of plant parts, but to identify active species and suggest potential compounds within the specific EOs that were use that may underly their sprout suppressive abilities.

  1. The authors are not satisfied with the legend representation of the data. For example, in Tables 2 and 3, it would be better to change Treatment vs Control to Plant EO, and it would be better to change 30, 45... days of X axis to Storage period for readability. Also Fig. In 2, you need to change Number to Number of sprouts and Time to Storage period.

Response: Tables 2, 3, and 7 have been altered to include “Plant EO” and “Duration of Storage” on the x-axis as suggested. The labels of Figures 1 and 2 have been changed from “Length” to “Sprout Length (mm)”, “Number” to “Number of Sprouts”, and “Time” to “Duration of Storage”.

And scientific names are in italics. need to be corrected.

Response: Scientific names in Figures 1 and 2 are now in italics.

  1. There is no explanation as to why the storage period is set to 90 days. It is unknown whether the authors confirmed the fumigation effect of EO for 90 days prior to this experiment. As shown in Figure 1, after 45 days, the growth of the shoots accelerates rapidly. In addition, the treatment concentration of EO should be performed as a prior study. The duration of the experiment may have been short.

Response: The storage period was set to 90 days due to limitations preventing longer storage durations. When commercial EO sprout suppressants are used they are often applied on a semi-weekly to monthly basis. As only a single application of EO was used in the present study, we believe a storage length of 90 days is sufficiently long to observe meaningful effects on tuber sprouting. Future studies could investigate longer storage lengths or may test different concentrations of effective EOs.

  1. The authors performed GC-MS analysis of the three essential oils with the best inhibitory effect. Therefore, α-pinene was found in M. communis EO, eucalyptol in M. quinquenervia, and geranial in C. citratus. However, there is no explanation as to how these components affect shoot inhibition (including previous studies). That is, the biggest problem of this paper is that there is no consideration on the germination inhibitory effect of the EO component. For example, these EO components may have factors such as inhibition of biosynthesis of hormones such as auxin, increase of biosynthesis of hormones such as ABA, and changes in energy sources such as reducing sugars. Although this experiment has not been conducted, it may be considered through previous studies.

Response: Greater discussion of previous studies investigating the effects of citral, eucalyptol, and pinene on plant hormone levels, germination, and plant growth has been added (Lines 123 – 125, 199 – 212).

  1. Line 168-187 does not need to be mentioned. This is because the three types of EO revealed that there is significance, and the story is mainly developed based on this.

Response: These lines have been removed.

  1. I don't know why 3 SEM pictures are listed in Figure 4. The authors would like to show only a picture of the control and processing section, or a characteristic part.

Response: As C. citratus EO treatment resulted in complete suppression of sprouting, it is unknown whether the EO achieved this response by damaging the bud tissue, by altering the hormonal balance within the tuber flesh, or through another mechanism. We believe the SEM images show the presence of healthy, undeveloped bud tissue on tubers treated with C. citratus EO, suggesting that this treatment may possess a mode of action that is different from the physical damage typically associated with EO sprout suppressants.

  1. Controlling potato tubers in a controlled environment (even if tuber sprouting is suppressed) may cause changes in the tissue and nutritional quality of potatoes, and may reduce tuber weight. In other words, the authors should specify the limitations of the study.

Response: A few sentences have been added to address the limitations of the study and to suggest areas of future research regarding EOs’ effects on potato flavor, texture, and nutritional quality (Lines 230 – 234).

Reviewer 2 Report

This MS found that EO could suppress potato sprout. The EOs of Myrtus communis and Melaleuca quinquenervia significantly reduced sprout length relative to the control but did not have any effect on sprout number. These findings demonstrate the potential of select EOs as effective potato sprout suppressants that could replace CIPC use in this industry. Below are the comments.

1.        Figure 1, 2, the different groups in the figure are not very clearly differently from each other.

2.        The mechanism of EO in suppressing potato sprout is not clear. The genes related to sprout may be determined.

3.        Table 4. M. communis EO constituents identified via GC—MS—FID analysis. Could the concentration of compound be provided?

4. Will EO affect the inner part of potato?

Author Response

This MS found that EO could suppress potato sprout. The EOs of Myrtus communis and Melaleuca quinquenervia significantly reduced sprout length relative to the control but did not have any effect on sprout number. These findings demonstrate the potential of select EOs as effective potato sprout suppressants that could replace CIPC use in this industry. Below are the comments.

  1. Figure 1, 2, the different groups in the figure are not very clearly differently from each other.

Response: The black and white versions of Figures 1 and 2 have been replaced with color versions to see differences between groups more easily.

  1. The mechanism of EO in suppressing potato sprout is not clear. The genes related to sprout may be determined.

Response: This is a valuable point. It is possible that EOs may affect the expression of genes related to dormancy in potato tubers or may alter hormone concentrations within the tuber flesh and skin to prolong dormancy. The importance of proteomic analysis has been incorporated into the text (Lines 218-228).

  1. Table 4. M. communis EO constituents identified via GC—MS—FID analysis. Could the concentration of compound be provided?

Response: The area percentages given in Tables 4, 5, and 6 represent the relative proportions of each compound to all other constituents in the FID chromatogram. The goal of this analysis was not to determine concentrations of constituents, but to determine their relative proportions which may provide clues as to which compound(s) may be active ingredients involved in sprout suppression.

  1. Will EO affect the inner part of potato?

Response: This is a great question. While the present study was focused solely on the effects of EOs on potato sprouting, future studies may elucidate their effects on other potato quality measures such as glucose/sucrose concentrations, texture (puncture strength), or cooked potato product flavor.

Reviewer 3 Report

Potatoes are a staple food in the diet of millions of people and the constant demand requires the storage of large quantities to meet consumption throughout the year, therefore potato sprouting during storage is a major problem leading to loss of income and waste of food. As bans on common synthetic suppressants become more widespread, more attention is turning to organic alternatives.

The authors of the manuscript entitled “Evaluation of essential oils as sprout suppressants for potato (Solanum tuberosum) at room temperature storage” present a study on potato (Solanum tuberosum) at room temperature storage. The study is of interest taking into account the high demand for food that has not been acted upon with chemical substances such as chlorpropham (CIPC).  The authors evaluate the effects of ten essential oils (Myrtus communis, Melaleuca quinquenervia, Myristica fragrans, Commiphora erythraea, Citrus aurantium, Citrus sinensis, Bursera graveolens, Petroselinum sativum, Pogostemon cablin, Cymbopogon citratus) as potential sprout suppressants in stored potatoes at room temperature.

Of these essential oils, their results identify that Cymbopogon citratus essential oil is a highly effective germination suppressant that completely suppresses germination in Ranger Russet potatoes for up to 90 days at room temperature. Also, in agreement with their results, it is also observed that the efficacy of Myrtus communis and Melaleuca quinquenervia essential oils could be improved if they are applied repeatedly during storage. According to the results presented by the authors in this manuscript, essential oils can be considered as an organic alternative to present practices dependent on the use of CIPC.

In this situation, I recommend that the article be published in the current form.

Author Response

Potatoes are a staple food in the diet of millions of people and the constant demand requires the storage of large quantities to meet consumption throughout the year, therefore potato sprouting during storage is a major problem leading to loss of income and waste of food. As bans on common synthetic suppressants become more widespread, more attention is turning to organic alternatives.

The authors of the manuscript entitled “Evaluation of essential oils as sprout suppressants for potato (Solanum tuberosum) at room temperature storage” present a study on potato (Solanum tuberosum) at room temperature storage. The study is of interest taking into account the high demand for food that has not been acted upon with chemical substances such as chlorpropham (CIPC).  The authors evaluate the effects of ten essential oils (Myrtus communis, Melaleuca quinquenervia, Myristica fragrans, Commiphora erythraea, Citrus aurantium, Citrus sinensis, Bursera graveolens, Petroselinum sativum, Pogostemon cablin, Cymbopogon citratus) as potential sprout suppressants in stored potatoes at room temperature.

Of these essential oils, their results identify that Cymbopogon citratus essential oil is a highly effective germination suppressant that completely suppresses germination in Ranger Russet potatoes for up to 90 days at room temperature. Also, in agreement with their results, it is also observed that the efficacy of Myrtus communis and Melaleuca quinquenervia essential oils could be improved if they are applied repeatedly during storage. According to the results presented by the authors in this manuscript, essential oils can be considered as an organic alternative to present practices dependent on the use of CIPC.

In this situation, I recommend that the article be published in the current form.

Response: Thank you.

Round 2

Reviewer 1 Report

The authors have made corrections to the review. However, the review nature has not yet been supplemented. The authors gave a description of the reviewer. However, the reviewer is to promote scientific development by reading the results of this study to many readers. In this regard, it is considered that more corrections should be made.

The answer to the use of commercial commercial essential oils is lacking. At least the limitations of the study should be presented.

2. Information on the 90-day storage period: The answer has been answered, but this should be added to the thesis.

3. Describe the mechanism of inhibition of potato tuber sprouting for the main monoterpenes: The authors described the mechanism of inhibition of tuber sprouting for citral, eucalyptol, and pinene, but in this study, these substances (α-pinene, eucalyptol, and geranial) are not referenced. There doesn't seem to be anything at all.

AMERICAN POTATO JOURNAL (Vol. 68 1991) VAUGHN AND SPENCER: VOLATILE MONOTERPENE ... etc.

Regarding the explanation of SEM picture No. 4. The author's answer is not just the reviewer's answer. It should be stated in the manuscript

Author Response

The authors have made corrections to the review. However, the review nature has not yet been supplemented. The authors gave a description of the reviewer. However, the reviewer is to promote scientific development by reading the results of this study to many readers. In this regard, it is considered that more corrections should be made.

The answer to the use of commercial commercial essential oils is lacking. At least the limitations of the study should be presented.

Response: Additional information regarding the plant parts used, their origin, and the method of distillation have been included in Table 8 in section 3.2 Plant Materials of the Methods section (Lines 309 – 312). This information was obtained from the suppliers’ websites. The limitations of the study regarding the plant parts used for extraction have been included in the manuscript and future areas of research are suggested (Lines 264 – 274).

  1. Information on the 90-day storage period: The answer has been answered, but this should be added to the thesis.

Response: Information regarding the choice of a 90-day storage period has been added, and future areas of research are suggested (Lines 275-283).

  1. Describe the mechanism of inhibition of potato tuber sprouting for the main monoterpenes: The authors described the mechanism of inhibition of tuber sprouting for citral, eucalyptol, and pinene, but in this study, these substances (α-pinene, eucalyptol, and geranial) are not referenced. There doesn't seem to be anything at all.

Response: The source provided below by the reviewer discussing pure applications of eucalyptol, α-pinene, and citral, an aldehyde mixture of geranial and neral, has been incorporated into the discussion (Lines 212 – 215). These pure compounds have been associated with varying degrees of necrosis of developing sprout tissue. As necrosis was not observed due to C. citratus EO treatment in the present study, this suggests a different mode of action of citral, such as alteration of the hormonal balance within potato which has been suggested by Food Chemistry (Vol. 26 2022) Huang et al.: Emulsification-based interfacial… (Lines 225 – 226). The effect of eucalyptol and α-pinene on corn mitochondrial metabolism is also mentioned as another potential mode of action by J Chem Ecol (Vol. 26 2000) Abrahim et al.: Effects of four monoterpenes on germination… (Lines 227 – 230). While the present study did not investigate these pure compounds as sprout suppressants, this is suggested as an area of future research (Lines 232 – 238).

AMERICAN POTATO JOURNAL (Vol. 68 1991) VAUGHN AND SPENCER: VOLATILE MONOTERPENE ... etc.

Response: This resource has been incorporated into the discussion (Lines 212 – 215).

Regarding the explanation of SEM picture No. 4. The author's answer is not just the reviewer's answer. It should be stated in the manuscript

Response: The explanation regarding the SEM pictures has been provided in the discussion (Lines 215 – 223).